# Diagnostic Accuracy of Machine-Learning Models on Predicting Chemo-Brain in Breast Cancer Survivors Previously Treated with Chemotherapy: A Meta-Analysis

**DOI:** 10.3390/ijerph192416832

**Published:** 2022-12-15

**Authors:** Adina Turcu-Stiolica, Maria Bogdan, Elena Adriana Dumitrescu, Daniela Luminita Zob, Victor Gheorman, Madalina Aldea, Venera Cristina Dinescu, Mihaela-Simona Subtirelu, Dana-Lucia Stanculeanu, Daniel Sur, Cristian Virgil Lungulescu

**Affiliations:** 1Department of Pharmacoeconomics, University of Medicine and Pharmacy of Craiova, 200349 Craiova, Romania; 2Department of Pharmacology, University of Medicine and Pharmacy of Craiova, 200349 Craiova, Romania; 3Department of Oncology, “Carol Davila” University of Medicine and Pharmacy, 050474 Bucharest, Romania; 4Institute of Oncology, Prof Dr. Alexandru Trestioreanu, Soseaua Fundeni, 022328 Bucharest, Romania; 5Department of Psychiatry, University of Medicine and Pharmacy of Craiova, 200349 Craiova, Romania; 6Department of Health Promotion and Occupational Medicine, University of Medicine and Pharmacy of Craiova, 200349 Craiova, Romania; 711th Department of Medical Oncology, University of Medicine and Pharmacy “Iuliu Hatieganu”, 400125 Cluj-Napoca, Romania; 8Department of Oncology, University of Medicine and Pharmacy Craiova, 200349 Craiova, Romania

**Keywords:** chemo-brain, machine learning, diagnostic accuracy, breast cancer, chemotherapy

## Abstract

We performed a meta-analysis of chemo-brain diagnostic, pooling sensitivities, and specificities in order to assess the accuracy of a machine-learning (ML) algorithm in breast cancer survivors previously treated with chemotherapy. We searched PubMed, Web of Science, and Scopus for eligible articles before 30 September 2022. We identified three eligible studies from which we extracted seven ML algorithms. For our data, the χ^2^ tests demonstrated the homogeneity of the sensitivity’s models (χ^2^ = 7.6987, df = 6, *p*-value = 0.261) and the specificities of the ML models (χ^2^ = 3.0151, df = 6, *p*-value = 0.807). The pooled area under the curve (AUC) for the overall ML models in this study was 0.914 (95%CI: 0.891–0.939) and partial AUC (restricted to observed false positive rates and normalized) was 0.844 (95%CI: 0.80–0.889). Additionally, the pooled sensitivity and pooled specificity values were 0.81 (95% CI: 0.75–0.86) and 0.82 (95% CI: 0.76–0.86), respectively. From all included ML models, support vector machine demonstrated the best test performance. ML models represent a promising, reliable modality for chemo-brain prediction in breast cancer survivors previously treated with chemotherapy, demonstrating high accuracy.

## 1. Introduction

Breast cancer (BC) is the leading cancer among women worldwide [1]. Currently, approximately 80% of patients with BC are individuals aged > 50. Survival depends on both stage and molecular subtypes [2]. The diagnosis and treatment of breast cancer may have a detrimental impact on the physical and emotional well-being of women because of the adverse effects of treatment, fear of death, and feelings of social devaluation [3]. The mortality of BC has decreased in recent decades because of the advanced therapies and better management of the personalized risk profile of every patient. Benefiting from these techniques and therapies, the paradigm shift will be toward reducing the negative consequences of oncological treatment for improved quality of life in breast cancer patients. Among these treatments, chemotherapy remains an important milestone in the BC therapy approach. Breast cancer survivors who have undergone chemotherapy might complain about cognitive-induced cognitive impairment (CICI) during and after treatment. CICI or “chemo-brain” affects more than 50% of BC patients and is used to describe the changes in cognitive function following systemic chemotherapy [4]. The impacted cognitive functions include executive function, memory, psychomotor function, attention, visual-spatial skills, processing speed, and reaction time [5,6].

As medical technology continues to advance, the number of breast cancer survivors is increasing and the social and economic burden of CICI is increasing. Therefore, early detection of the occurrence of CICI is crucial for early clinical intervention and the prognosis of patients. Chemo-brain is frequently assessed using neuropsychological testing, but there is a lack of an adequate statistical and technical threshold to distinguish between impaired and non-impaired cognitive function [7,8]. Numerous neuroimaging investigations have demonstrated that CICI is a brain illness characterized by the extensive disruption of large-scale neural networks [9,10,11]. However, the variability of the CRCI has not been sufficiently studied, as the majority of studies have explored it binarily, and we believe that neuroimaging has the potential for diagnosing chemo-brain. To accomplish this purpose, we performed a meta-analysis of chemo-brain diagnostic in order to assess the accuracy of a machine-learning algorithm in breast cancer survivors previously treated with chemotherapy.

Several neuroimaging investigations [11,12] have demonstrated that CICI is characterized by a reduction in the volume and density of gray matter and destruction of the structural integrity of white matter. In addition, resting-state functional magnetic resonance imaging (rs-fMRI) investigations indicated that breast cancer patients with CICI exhibited substantial abnormalities of local functional activities and connections [13]. Rs-fMRI is a safe and noninvasive method to investigate functional and structural alterations in the human brain, during which patients are required to relax and clear their minds throughout the MRI scan without following any instructions, making the experiment simpler and more reproducible in clinical settings. Dumas et al. [14] showed decreased functional connectivity in the default mode network (DMN) one month and one year after chemotherapy in breast cancer patients. These findings suggest a detrimental effect of chemotherapy on brain functional connectivity that is potentially related to subjective cognitive assessment. However, the majority of this research is conducted independently according to brain functional connectivity or regional activity. Given the multimodal aspects of brain function, it will be difficult to establish the differentiating traits and apply them to the individual categorization of CICI.

To resolve these issues, classification models using rs-fMRI and machine learning have been used for the accurate early detection of CICI. There are various machine-learning models, e.g., logistic regression (LR), decision tree classifier (CART), XGBoost (XGB), deep learning (DL), support vector machine (SVM), and random forest (RF), with known efficacy in the detection the subtle alterations of the brain in postchemotherapy BC patients [15,16,17,18,19,20]. LR is a typical linear model based on the logistic function for classification rather than a regression model, with higher training and prediction speed, good accuracy for many simple data sets and it performs well when the dataset is linearly separable. LR is less inclined to over-fitting and can interpret model coefficients as indicators of feature importance, but it remains difficult to obtain complex relationships using LR. More compact algorithms such as neural networks can outperform this algorithm [21]. SVMs are easily interpretable with an efficient classification, which enhances the predictive accuracy of health data, but they are not efficient in handling large sets of datasets. The target classes in a large dataset usually overlap, affecting the classification and predictability. However, SVM needs meticulous calibration and preprocessing [22].

XGB is very effective, adaptable, and accurate. It is built on the gradient boosting architecture and has demonstrated its excellent performance in numerous machine learning competitions [23]. In XGB, weights play a crucial role. All independent variables are assigned weights and subsequently fed into the decision tree used to predict outcomes. The weight of variables for which the tree made incorrect predictions are increased, and these variables are then given to a second decision tree. The ensemble of these independent classifiers/predictors generates a robust and more accurate model [24].

CART is a variation of the decision tree algorithm and can handle both classification and regression tasks. CART is a promising model that employs an if/then question-based machine learning algorithm, and it has been widely used in recent years [25]. The model is straightforward to visualize, feature scaling is unnecessary, requires minimal supervision, and produces easy-to-understand models. These are three of CART’s primary advantages. However, CART has a propensity for overfitting, high variance, and poor induction performance [24,26].

Random forest is an ensemble technique that uses many decision trees and the Bootstrap and Aggregation process, also known as bagging, to perform regression and classification tasks. Rather than depending on individual decision trees to determine the ultimate result, this method combines many decision trees [27]. Random forest employs numerous decision trees as its primary learning models, combats the overfitting that frequently occurs in CART, and has many benefits over CART. Despite this, RF has a more difficult visualization and is unsuitable for high-dimensional and sparse data sets [24].

In recent years, the evolution of image interpretation systems that use machine learning (ML) approaches have been rapid. The next obvious step was to develop intelligent computers that could independently learn and extract picture comprehension characteristics. CNN (convolutional neural network), recurrent neural network, and generative adversative networks are examples of the techniques used by deep learning. One such model is the CNN model, which automatically learns and extracts the required characteristics for medical picture comprehension [28]. CNN models have been characterized as black boxes, and much research is being conducted to analyze output at every layer. Due to the involvement of medical imaging, we want an effective prediction system that is also capable of communicating a judgment. Picture captioning is another area of study. This will allow doctors to comprehend the network’s perspective at both the output and intermediate layers [29].

The purpose of this research was to assess which machine-learning models are the most effective and accurate at recognizing chemo-brains and to demonstrate that there are valid machine-learning perspectives that can distinguish BC patients with CICI from healthy controls, thereby providing potential neuroimaging evidence for early diagnosis of the occurrence of chemo-brain and clinical intervention in breast cancer patients. In the future, we anticipate using machine learning as a paradigm for clinical CICI monitoring.

## 2. Materials and Methods

### 2.1. Overview

This review was conducted according to the PRISMA-DTA [30], the checklist being presented in the Appendix A. 

### 2.2. Literature Search and Study Selection

A search in PubMed, Web of Science, and Scopus, was performed until 30 September 2022, using the keywords: (“machine learning” OR “deep learning”) AND “breast cancer” AND (“chemo-brain” OR “cognitive impairment” OR “chemotherapy-related cognitive impairment” OR “CRCI” OR “chemotherapy-induced cognitive impairment” OR “CICI”) AND “chemotherapy” AND “MRI”. The review of the reference lists from the included studies was also performed.

As guidelines require, two reviewers (M.B. and M.-S.S.) selected potentially relevant studies and disagreements were resolved by a third reviewer (C.V.L.). All the results were imported into Rayyan, where duplicate papers were automatically screened and deleted [31].

We selected the articles based on the following criteria: (1) articles female post-chemotherapy breast cancer survivors; (2) including groups with chemo-brain and healthy controls; (3) articles that used machine learning models or algorithms; (4) articles that used magnetic resonance imaging (MRI) data; (5) articles written in English; (6) articles that reported the performances of confusion matrix. We excluded: (1) articles that did not reported sufficient data; (2) the publication was a conference abstract, or a review article.

A spreadsheet was used to extract the study information: name, country of the data, patient groups, algorithms, MRI manufacturer, outcomes of ML detection (from which to extract TP, true positive; FP, false positive; FN, false negative; TN, true negative). Additionally, all the models from an article were extracted if they used different classifiers or prediction classes. Otherwise, if the articles presented chemo–brain relatively similar models, we extracted the model with the highest accuracy.

### 2.3. Quality Assessment

We used the Quality Assessment of Diagnostic Accuracy Studies (QUADAS-2) tool [32] to evaluate the quality of the study that were included in our meta-analysis. The overall quality was considered very poor, poor, moderate, and good, depending on the score calculated after rating the four domains: patient selection, index test, reference standard, and flow and timing. All domains are assessed in terms of risk of bias, whereas the first 3 domains are evaluated from the point of view of concerns regarding applicability. We assigned the values of −1, 0, and 1 to low, unclear, and high and the total score could range from −7 to 7.

### 2.4. Statistical Analysis

Two R packages were used for carrying out the statistical analysis: mada [33] and metafor [34]. Each ML model was summarized by the pooled diagnostic odds ratio (DOR), sensitivity and specificity together with their 95% confidence intervals. The correlation of sensitivities and false positive rates was calculated to give a hint if the cut-off value problem was present. As sample size was too small, the univariate approach to the meta-analysis of diagnostic accuracy was approached. Pooling sensitivities and specificities were performed, as well as the positive and negative likelihood ratios, and θ, the accuracy parameter of the proportional hazards model for diagnostic meta-analysis [35]. Cochran’s Q and Higgins’ I^2^ were calculated as measures of heterogeneity [36].

For a more informative graphical form, we established crosshairs plots [37], which are a combination of both receiver operating characteristic (ROC) curve and forest plot, demonstrating the bivariate relationship, the degree of heterogeneity (the crosshairs are wider with increased sample size). The ROCellipse plots confidence regions which evidence the uncertainty of the pair (sensitivity, false positive rate) as ellipses on logit ROC space.

## 3. Results

### 3.1. Eligible Studies and Quality Assessment

As presented in Figure 1, three research articles were identified in the three databases. After removal of 14 duplicates, the remaining 20 articles were included in the screening process. Finally, three studies containing seven machine learning models were included in our study.

Figure 2 shows the quality assessment of the three included studies using the Quality Assessment of Diagnostic Accuracy Studies QUADAS-2 tool. The studies avoided inappropriate inclusions or exclusions of the patients, with no existence of flow and timing risk of bias. Only one study [17] was considered unclear regarding the patient selection on consecutive or randomized sample. Only one study [15] was considered unclear regarding the risk of bias for reference standard and blinding to index test result.

### 3.2. Eligible Studies and Their Characteristics

The main characteristics of the included studies are summarized in Table 1. Four models were included from Chen et al. [17], two models from Lin et al. [15], and one model from Wang et al. [16]. The studies were conducted in China [16] and Taiwan [15,17]. In this systematic review and meta-analysis, 114 cancer patients and 119 healthy controls were included.

### 3.3. Descriptive Statistics

The study with the highest accuracy was the latest published in 2022 (91.9%), using a SVM classifier on seventeen features. The sensitivity and specificity together with their confidence intervals of the eligible models that were described in Table 1 are presented in Figure 3. Sensitivity values for ML models ranged between 0.68 (95% CI: 0.46–0.85) and 0.92 (95% CI: 0.80–0.97), while specificity values ranged between 0.80 (95% CI: 0.58–0.92) and 0.85 (95% CI: 0.64–0.95). Testing for the equality of sensitivities demonstrated the homogeneity of the models (χ^2^ = 7.6987, df = 6, *p*-value = 0.261). No differences were observed between the specificities of the ML models (χ^2^ = 3.0151, df = 6, *p*-value = 0.807). 

Apart from these univariate analysis, Figure 3 shows the study on ROC space.

The plots in Figure 4 confirmed this homogeneity in ROC space. We could visually evaluate the points for the specificity and sensitivity paired within the seven models, all of them being situated at the left of the main diagonal of the square graph.

In addition to the confidence intervals, Cochran’s Q statistic and Higgins’ I^2^ were calculated: 6.192 (df = 6, *p*-value = 0.402) and, respectively, 3.103%. The coefficient θ (theta) was 0.093 (95% CI: 0.065–0.123). The smaller this diagnostic accuracy parameter, θ, the larger the area under the ROC curve and thus the more accurate the ML diagnostic test. The values for DOR (95% CI) and τ^2^ (95% CI) were 20.38 (11.92–34.69) and 0.07 (0.0–2.9), respectively. The log DOR value of each model ranged between 2.16 (95% CI: 0.70–3.62) and 4.85 (95% CI: 3.18–6.52). Apart from these univariate analysis, Figure 5 shows the forest plot of the log DOR values together with the summary estimation. As pooled log DOR was 3.01 (95% CI: 2.48–3.55), the diagnostic odds ratio had more than 100 values indicating very good test performance. Model 2 had the worst test performance, whereas model 7 had the best test performance.

### 3.4. Overall Model

The correlation of sensitivities and false positive rates is low (rho = −0.207, 95% CI: −0.830 to 0.647). Based on the HSROC curve plot (Figure 6), there was a small deviation of the individual models from the curve. 

The pooled area under the curve (AUC) for the overall ML models in this study was 0.914 (95% CI: 0.891–0.939) and partial AUC (restricted to observed false positive rates and normalized) was 0.844 (95% CI: 0.80–0.889). Additionally, the pooled sensitivity and pooled specificity values were 0.81 (95% CI: 0.75–0.86) and 0.82 (95% CI: 0.76–0.86), respectively.

## 4. Discussion

With so much emphasis on the development of artificial intelligence (AI), there is a rising interest in its use in cancer detection, diagnosis, and treatment side effects. In the area of medical imaging, decision-making-capable machine-learning algorithms are required. Complex and subtle variations that cannot be immediately identified by doctors may now be recognized by cutting-edge machine learning technology. Cognitive impairment is a well-known side effect of cancer and its therapies, although it is still poorly understood due in part to the absence of a defined definition or diagnostic criteria. CICI has been studied in the past as a dichotomous condition in which people are either impaired or not impaired, which is unlikely to be realistic given the complexity of the cognitive function.

In this current meta-analysis, we found that ML algorithms may be used for the diagnosis of chemo-brain in breast cancer patients previously treated with chemotherapy, with good diagnostic accuracy, in terms of sensitivity and specificity.

We have analyzed three types of ML classes: SVM [16], LR [17], and deep learning [15], containing seven machine learning models: LR-GFA standardized and unstandardized, LR-mReHo standardized and unstandardized [17], SE-ResNet-50 [15], SE-ResNet-121 [15], and SVM-model [16].

Contrary to popular belief, logistic regression is a regression model. Logistic regression becomes a classification technique only when a decision threshold is brought into the picture. The setting of the threshold value is a very important aspect of logistic regression and is dependent on the classification problem itself. LR models do not require the selection of a learning rate; they frequently run faster and can numerically approximate the gradient. However, LR is more complex and, unless the specifics are learned, more of a black box [21].

When it comes to deep learning, the convolutional neural network (CNN) performs the best in image recognition [15,16]. In a CNN, there are three types of layers: input layers, in which we give input to our model; hidden layer, where the input is fed; and output layer, where the output from the hidden layers is then fed into a logistic function, which converts the output of each class into the probability score of each class. The data are then fed into the model, and the output from each layer is called feedforward [24]. As the name suggests, it is a neural network that makes use of convolutional operations to classify and predict. Some of the advantages of CNN are weight sharing, memory saving, and equivariance, the property of CNNs whereby, upon a change in the input, a similar change is reflected in the output; this helps identify any drastic change in the output and retain the reliability of the model [28]. There are also independents of transformation and independents of local variations in image. There are also other types of neural networks in deep learning, but for identifying and recognizing objects and images, CNNs are the network architecture of choice [15,29].

SVM is a supervised machine learning algorithm that can be used for both classification and regression challenges. However, it is mostly used in classification problems and works well with unstructured and semi-structured data such as images. In practice, SVMs models scale well to high-dimensional data and have generalization; the risk of over-fitting is lower in this algorithm [24]. The kernel trick is the real strength of SVM, but choosing an appropriate kernel function is not easy. A kernel trick is a simple method for projecting data from a non-linearly separable training set into a higher-dimensional space where it becomes linearly separable. Thus, choosing the right kernel function and regularization is of great importance [22].

The most accurate research was the latest published in 2022 using an SVM classifier on seventeen features (91.9% accuracy). A recent study [15] developed the support vector machine (SVM) with patterns of DMN connectivity, which could discriminate between chemotherapy-treated BC survivors and non-chemotherapy-treated BC survivors with 90–91% accuracy, while disregarding the effect of other parameters of rs-fMRI functional connection and activity on breast cancer patients after chemotherapy. As noted earlier, SVM is only effective when dealing with limited dataset sets, such as in the research by Wang et al. [16] (89 BC patients and 34 HC). In a large dataset, the target classes often overlap, impacting classification and prediction. In addition, SVMs need rigorous calibration and preprocessing, and their application is not simple [22]. 

On the other hand, this study was the only one that could identify breast cancer patients with subjective cognitive complaints related to chemotherapy from BC patients before chemotherapy, not just from HCs like the other ML algorithms [15,17] analyzed. In this research, they also incorporated the multi-level rs-fMRI features comprising brain functional activity, local functional activity: low-frequency fluctuation (ALFF), fractional ALFF (fALFF), and regional homogeneity (ReHo), and graph theory analysis, which were selected by t-test, removal of high pairwise correlation, and least absolute shrinkage and selection operator (LASSO) regression to construct the linear SVM model. In comparison, in the work of Chen et al. [17], they used generalized q-sampling imaging (GQI), voxel wise analysis, regional summation, and several machine-learning models: LR mean regional homogeneity (LR-mReHo) and LR generalized fractional anisotropy (LR-GFA) to demonstrate the efficacy of these techniques. They also preprocessed the MRI images to optimize the prediction s model, by using regional summation to reduce the dimension of features. In addition, they attempted additional feature selection or reduction techniques, such as principal component analysis (PCA), variance threshold, and choosing k best, among others. Even if these feature selection techniques harmed their findings, they may still be applicable for developing alternative machine-learning models. Regarding preprocessing, their findings indicate that standardization may improve the precision of LR on both GQI and rs-fMRI data sets. In overfitting, when the ratio of training samples to dimensionality is low, a significant difficulty often arises. This is the issue most MRI researchers face. During overfitting, a model tends to learn the particular pattern of tiny samples rather than the desired overall pattern. Leave-one-out cross-validation (LOOCV), which might decrease overfitting, is a suitable technique for a small data set with a lengthy training duration [38], because it allows models to exclude data that might cause overfitting. In addition, since the difference in size between the training sets used in each fold and the full data set is just a single pattern, LOOCV is approximately unbiased, maintaining its great reliability.

Moreover, we can state that the selection of a model and preprocessing may have a significant impact on the classifications produced by machine learning. Given that small data sets often require less time to analyze, we recommend that researchers with a small quantity of data attempt analyzing their data using a variety of models to discover the particularly fitted one.

Heterogeneity, which is common in diagnostic meta-analyses, is the result of variations among the different included studies [39]. These variations mainly include differences in the study population, study design, interventions, and interpretations of results. In our case, the heterogeneity between studies was not significant, regarding the MRI acquisition, all MRI data were collected on Siemens Medical Solution Scanners from Erlangen Germany, one3.0- Tesla Trio TIM Scanner [16], one Verio Siemens Scanner [15] and one Magnetom Aera Scanner [17]. Unfortunately, none of the studies analyzed MRI scans pre- and post-chemotherapy from the same patients; they only compared MRIs from BC patients with HC. As a future direction, we propose analyzing images from the same patient before and after chemotherapy to improve accuracy.

In addition, in the three included studies [15,16,17], there was not even one external validation, which means testing the model with an out-of-sample dataset from one or more other centers. All studies analyzed MRI images from a single institution. Since the purpose of validation is to investigate the performance within patients from a different population, it is necessary to obtain a new dataset from a distinct source. As a result, the model’s generalizability could not be assured in the absence of external validation, causing the results to be overestimated.

Although some prior research [40,41,42] showed that chemotherapeutic drugs could impact the whole brain, the evidence that chemotherapy affects the entire brain is limited, according to our integrated findings. Each brain area has a specialized purpose. Human cognition is related with several integrated areas, and our brain does not confine its activities to a single region but rather acts as a complex system, as shown by mounting data. CICI may signify a disturbance in the brain’s network after chemotherapy.

Three of our analyzed ML algorithms (SE-RES NET 50, SE-DENSE NET 121, and SVM) [15,16] indicate that the identified cerebral regions are very similar to the components of the DMN, particularly the sub-regions comprising the prefrontal cortex, posterior cingulate cortex, inferior parietal lobule, and lateral temporal cortex. These zones were discovered to be critical for internal orientation cognition, cognitive control, and self-reference cognitive processing [43], which are all associated with a range of neuropsychiatric diseases [44]. The DMN is enriched with high-degree hub areas, suggesting that the DMN regions may serve as relay stations for sharing information across the brain [45,46].

By assessing functional connectivity in resting-state fMRI, researchers detected altered connectivity in particular DMN areas and proposed that these regions were associated with attention and memory deficits after chemotherapy [44,47,48]. These findings suggest that aberrant functional alterations in DMN may be one of the most efficient indicators for identifying CICI in breast cancer.

The precise mechanisms of CICI are unclear, but it does include direct neurotoxic damage, a reduction in neurogenesis, white matter anomalies in the central nervous system, and neuroinflammation [49]. Chemotherapy threatens the anatomical composition of the brain by diminishing the white matter integrity of the prefrontal cortex. Evidence suggests that cytokine dysregulation damages prefrontal and temporal cortical synaptic networks, hippocampal volume, and brain metabolism [50]. In our research, chemotherapy regimens were heterogeneous. One study [15] did not take into account the chemotherapeutic agent type and dosage at all, and another [16] included an increased number of chemotherapeutic therapies: cyclophosphamide, doxorubicin, epirubicin, docetaxel, fluorouracil, and combinations. Only the method using the LR algorithm compares patients with the same treatment regimen for improved diagnosis accuracy [17]. It is known that docetaxel compromises the structural integrity of the cerebral cortex and peripheral neurons [51]. Both doxorubicin and cyclophosphamide induced behavioral impairments, although only the last was associated with inflammation caused by microglia [52]. Variable levels of cytokine concentrations and cognitive impairment severity are associated with distinct chemotherapy regimens and doses. Additionally, cognitive disorder development must be investigated.

Moreover, a heterogeneous element was noted regarding other therapies involved in the BC patient treatments, such as targeted therapies, radiotherapy, or hormone therapy. We cannot evaluate their contribution to CICI development or brain MRI alterations. Cognitive impairment is commonly reported in breast cancer patients both during and after cessation of treatment, and is likely triggered by multiple factors, such as endocrine therapy, the cancer itself, stress, and the hormonal changes resulting from menopause, amongst others [53].

The subjective and objective cognitive performance of breast cancer patients are currently measured by using neuropsychological cognitive tests, such as Functional Assessment of Cancer Therapy-Cognitive Function (FACT-Cog) [54], Montreal Cognitive Assessment [55], Clock drawing test [56], or Trail making test [57]. Only Wang et al. [16] performed a cognitive assessment, completed within the same day of MRI scanning, and there was no significant difference in objective cognitive scores between chemotherapy-treated BC patient and non-chemotherapy-treated BC patient groups, which correspond to the results of previous studies [4,58].

The main limitation to our meta-analysis was the small number of included studies (only three, but with seven extracted models) as there were not many studies using AI models to detect CICI in breast cancer patients. According to Jim et al. [59] the first research on cognitive functioning in breast cancer survivors after chemotherapy treatment was published in 1998, but the number of medical AI studies saw a marked increase after 2017 [60]. We were able to extract the data we need from all seven models and highlight that SVM had the highest specificity, sensitivity, and accuracy among the included models. Nevertheless, all the models are trained on a small dataset, and a larger dataset is required to achieve better model performance. Moreover, verifying the model through external data sets would be the research direction. Further analyzing MRI scans from the same patient’s pre- and post-chemotherapy could also be considered in the future.

## 5. Conclusions

ML models represent a promising, reliable modality for chemo-brain prediction in breast cancer survivors previously treated with chemotherapy, demonstrating high accuracy. In this review and meta-analysis, we also put forward some existing problems of design and reporting that algorithm developers should consider. Based on these promising preliminary results and further testing on a larger dataset, artificial intelligence-assisted models could become an important tool for the computer-aided prediction and diagnosis of CICI. We are hopeful that this study could help the establishment of a clinically available model to track chemo-brain in the future.

## Figures and Tables

**Figure 1 ijerph-19-16832-f001:**
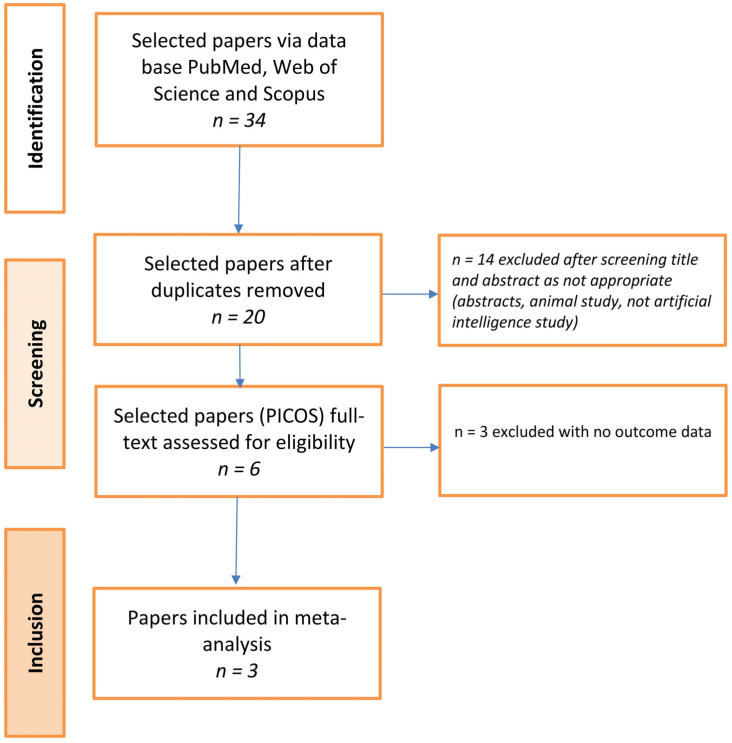
Flow diagram of the study selection process according to PRISMA guidelines.

**Figure 2 ijerph-19-16832-f002:**
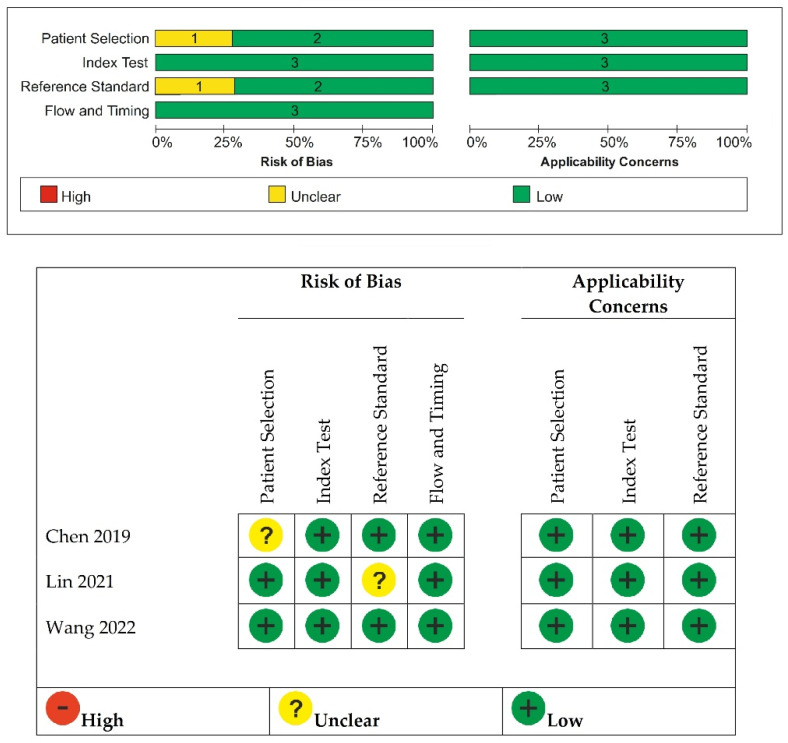
Methodologic quality of the included studies according to QUADAS-2 tool. Green represents low, yellow unclear and red high risk of bias (Lin 2021 [15], Wang 2022 [16], Chen 2019 [17]).

**Figure 3 ijerph-19-16832-f003:**
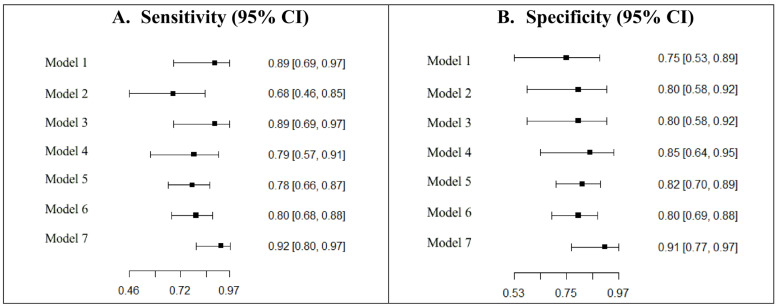
Sensitivity and specificity of machine learning models in the study. The names of the models are used accordingly to the models detailed in Table 1.

**Figure 4 ijerph-19-16832-f004:**
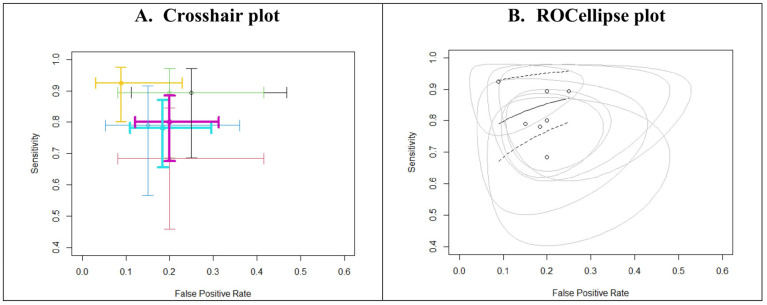
(**A**) Weighted crosshair plot with arbitrary coloring. (**B**) Summary plot with the proportional hazard model approach (PHM). The crosshairs displayed the uncertainty in sensitivity and specificity for every model. The solid line represents SROC curve, plotted together with the dotted lines representing confidence interval in the summary. The circle lines represent the 95% confidence regions for the model estimates.

**Figure 5 ijerph-19-16832-f005:**
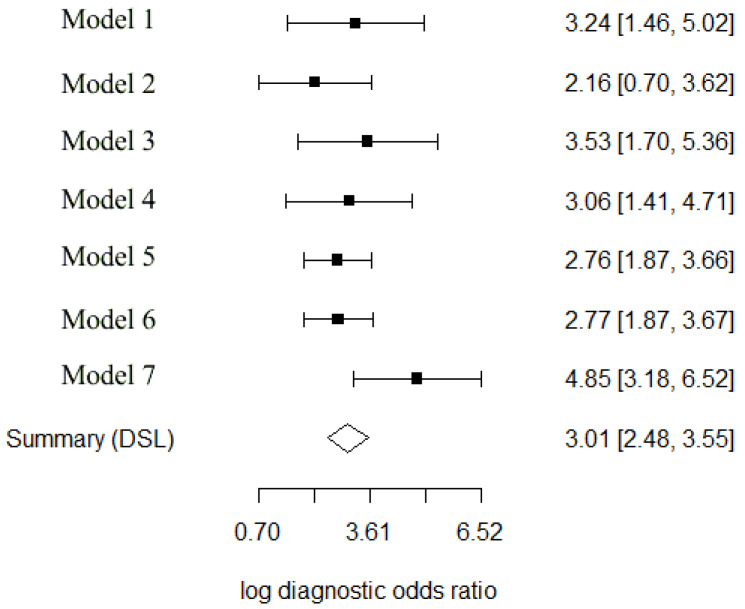
Forest plot for a univariate meta-analysis using the diagnostic odds ratio. The names of the models are used accordingly to the models detailed in Table 1.

**Figure 6 ijerph-19-16832-f006:**
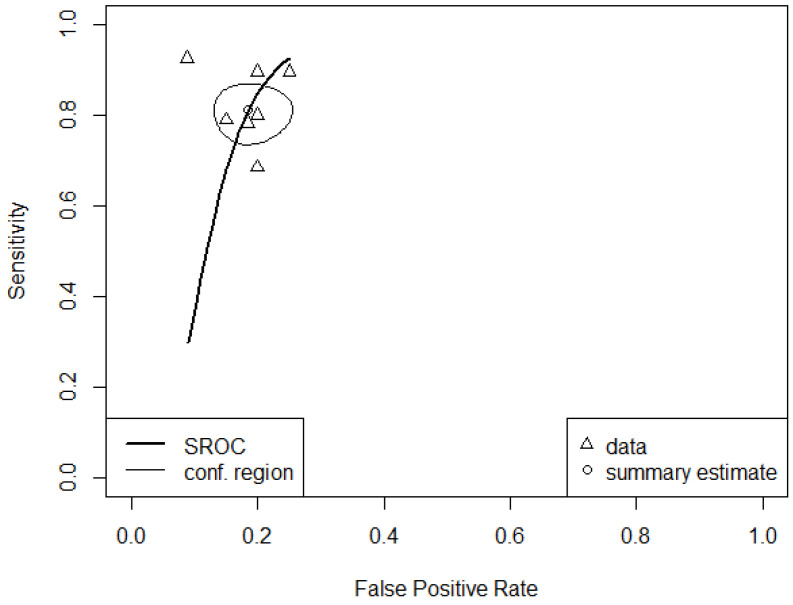
Hierarchical summary receiver operating characteristics—HSROC curve for overall machine learning models in the study.

**Table 1 ijerph-19-16832-t001:** The characteristics of the included studies.

Study	ID	Country	Patients Characteristics	Healthy Controls Characteristics	MRI Acquisition	Classifier
Chen 2019 [17]	1	Taiwan	N = 19 Mean age ± SD = 43.8 ± 6.4 Age range 32–55 Education 13.9 ± 2.2	N = 20 Mean age ± SD = 50.1 ± 2.5 Age range 43–55 Education 13.3 ± 2.3	Magnetom Aera; Siemens Medical Systems, Erlangen, Germany	LR-GFA standardized
	2					LR-GFA unstandardized
	3					LR-mReHo standardized
	4					LR-mReHo unstandardized
Lin 2021 [15]	5	Taiwan	N = 55 Mean age ± SD = 50 ± 8.09 Age range 32–65 Education 11.58 ± 3.8	N = 65 Mean age ± SD = 50 ± 8.09 Age range 31–67 Education 13.56 ± 3.1	Verio, Siemens, Germany	SE-ResNet-50
	6					SE-DenseNet-121
Wang 2022 [16]	7	China	N = 40 Mean age ± SD = 47.85 ± 6.87 Education 4.6 ± 3.6	N = 34 Mean age ± SD = 46.38 ± 9.88 Education 5.74 ± 3.12	Siemens Medical Solutions, Erlangen, Germany	SVM

SD, standard deviation; LR-GFA, linear regression-generalized fractional anisotropy; SE-squeeze and excitation; ResNet, residual neural network; DenseNet, dense convolutional network; SVM, support vector machine.

## Data Availability

Reported data are included in the manuscript.

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
