# Peer review of "Diagnostic Accuracy of Machine-Learning Models on Predicting Chemo-Brain in Breast Cancer Survivors Previously Treated with Chemotherapy: A Meta-Analysis"

_ijerph, 2022, doi:10.3390/ijerph192416832_

Round 1

Reviewer 1 Report

The work presents an important topic in the field of health, in addition, it is well presented, with a high technical level, a correct methodology for the selection of the work and its analysis. However, it can improve if you follow the recommendations indicated.

Recommendations and suggestions:

In the Abstract they must indicate the main contribution of their work to the state of the art.

In the introduction, when talking about ML models, the point is not complete. In addition, they should improve the closing of the section, giving a clearer idea of the contributions and the differences of their work against the state of the art.

It is recommended that more description be incorporated in section 2.3 about the use of QUADAS.

It is recommended that more description of section 3.1 be incorporated, about figure 2.

It is recommended that the names of the seven models in figure 4 be incorporated.

It is recommended that more description be incorporated about figures 5 and 6.

In the Discussion section, there needs to be more discussion about the studied ML models.

The paper highlights the importance of having a larger dataset, however, the authors do not give a clue about the value of its size.

Author Response

Thank you for your comments and suggestions concerning our manuscript entitled "Diagnostic Accuracy of Machine-Learning Models on Predicting Chemo-Brain in Breast Cancer Survivors Previously Treated with Chemotherapy: a Meta-Analysis"(Manuscript ID: ijerph-2068506). Those are all valuable and very helpful for revising and improving our paper. We have studied all comments carefully and have made corrections using tracked changes.

In the Abstract they must indicate the main contribution of their work to the state of the art.

We added more information about the main contribution.

In the introduction, when talking about ML models, the point is not complete.

We added more information about ML models:

XGB is very effective, adaptable, and accurate. It is built on the gradient boosting architecture and has demonstrated its excellent performance in numerous machine learning competitions [54]. In XGB, weights play a crucial role. All independent variables are assigned weights and subsequently fed into the decision tree used to predict outcomes. The weight of variables for which the tree made incorrect predictions are increased, and these variables are then given to a second decision tree. The ensemble of these independent classifiers/predictors generates a robust and more accurate model.

CART is a variation of the decision tree algorithm and can handle both classification and regression tasks. CART is a promising model that employs an if/then question-based machine learning algorithm, and it has been widely used in recent years. The model is straightforward to visualize, feature scaling is unnecessary, requires minimal supervision and produces easy-to-understand models; these are three of CART's primary advantages. However, CART has a propensity for overfitting, high variance, and poor induction performance.

Random Forest is an ensemble technique that uses many decision trees and the Bootstrap and Aggregation process, also known as bagging, to perform regression and classification tasks. Rather than depending on individual decision trees to determine the ultimate result, this method combines many decision trees. Random Forest employs numerous decision trees as its primary learning models, combats the overfitting that frequently occurs in CART, and has many benefits over CART. Despite this, RF has a more difficult visualization and is unsuitable for high-dimensional and sparse data sets.

In addition, they should improve the closing of the section, giving a clearer idea of the contributions and the differences of their work against the state of the art.

Thank you for this observation. We modified the last paragraph of the Introduction section:

 The purpose of this research was to assess which machine-learning models are the most effective and accurate at recognizing chemo-brains and to demonstrate that there are valid machine-learning perspectives that can distinguish BC patients with CICI from healthy controls, thereby providing potential neuroimaging evidence for early diagnosis of the occurrence of chemo-brain and clinical intervention in breast cancer patients. In the future, we anticipate using machine learning as a paradigm for clinical CICI monitoring.

It is recommended that more description be incorporated in section 2.3 about the use of QUADAS.

The use of QUADAS was more described.

It is recommended that more description of section 3.1 be incorporated, about figure 2.

 More description of Figure 2 was incorporated in section 3.1.

It is recommended that the names of the seven models in figure 4 be incorporated.

The names of the seven models were included in Figure 4 accordingly to Table 1. We added more details in the footer of Figure 4.

It is recommended that more description be incorporated about figures 5 and 6.

More description was incorporated about figures 5 and 6: see lines 242-258.

In the Discussion section, there needs to be more discussion about the studied ML models.

We agree and we added more discussion about the studied ML models:

We have analyzed three types of ML classes: SVM , LR , and deep learning, containing seven machine learning models: LR-GFA standardized and unstandardized, LR-mReHo standardized and unstandardized, SE-ResNet-50, SE-ResNet-121, and SVM-model.

Contrary to popular belief, logistic regression is a regression model. Logistic regression becomes a classification technique only when a decision threshold is brought into the picture. The setting of the threshold value is a very important aspect of logistic regression and is dependent on the classification problem itself. LR does not require the selection of a learning rate; they frequently run faster and can numerically approximate the gradient. However, LR is more complex and, unless the specifics are learned, more of a black box.

When it comes to deep learning, the convolutional neural network (CNN) performs the best in image recognition. In a CNN, there are three types of layers: input layers, in which we give input to our model; hidden layer, where the input is fed; and output layer, where the output from the hidden layers is then fed into a logistic function, which converts the output of each class into the probability score of each class. The data is then fed into the model, and the output from each layer is called feedforward. As the name suggests, it is a neural network that makes use of convolutional operations to classify and predict. Some of the advantages of CNN are weight sharing, memory saving, and equivariance, the property of CNNs whereby, upon a change in the input, a similar change is reflected in the output; this helps identify any drastic change in the output and retain the reliability of the model. There are also independents of transformation and independents of local variations in image. There are also other types of neural networks in deep learning, but for identifying and recognizing objects and images, CNNs are the network architecture of choice.

SVM is a supervised machine learning algorithm that can be used for both classification and regression challenges. However, it is mostly used in classification problems and works well with unstructured and semi-structured data like images. In practice, SVMs models scale well to high-dimensional data and have generalization; the risk of over-fitting is lower in this algorithm . The kernel trick is the real strength of SVM, but choosing an appropriate kernel function is not easy. A kernel trick is a simple method for projecting data from a non-linearly separable training set into a higher-dimensional space where it becomes linearly separable. Thus, choosing the right kernel function and regularization are of great importance.

The paper highlights the importance of having a larger dataset, however, the authors do not give a clue about the value of its size.

A meta-analysis could be performed on at least two studies or two models, in our case. In this systematic review and meta-analysis, 114 cancer patients and 119 healthy controls were included, one of the studies (Chen 2019) performed the ML on 19 patients with breast cancer and 20 with healthy controls. We did not define a limit for sample size when ML was applied in included studies because no research was published in this direction. There are a wide variety of ML models with different variables and parameters available, but some studies demonstrated the augmentation of training and testing sample sizes to enhance the overall accuracy [Ferro, A., Kotecha, S. & Fan, K. Machine learning in point-of-care automated classification of oral potentially malignant and malignant disorders: a systematic review and meta-analysis. Sci Rep 12, 13797 (2022). https://doi.org/10.1038/s41598-022-17489-1].

Reviewer 2 Report

This review article introduced automated meta-analysis to objectively to compare machine-learning models predicting chemo-brain caused by chemotherapy treating breast cancer in terms of accuracies and data quality. The method design is clear. The results of different machine learning methods were thoroughly compared with multiple statistical analyses. In the discussion sessions, the authors expanded their review on the impact of specific selected studies. They also discussed the possible mechanisms of the chemo-brain revealed by the current machine learning studies, the limitations of these models, and the future direction of development for applying machine learning to diagnose and study chemo-brain. I recommend publication of this short review if the following issues could be addressed.

1.     In the screening of the publications, 20 articles were input but only 3 were selected. Is it too restricted the screening criterion is? Are there any articles in the excluded ones worth discussions? Though they might lack some of the elements in the result reporting.

2.     In the literature searching step, the keyword “chemotherap” was used instead of “chemotherapy”. Is it a typo? Or “chemotherap” could function as the wildcard to pick all results with “chemotherapy” or “chemotherapeutic”? Please make double check and make sure this will not affect the expected searching results.

3.     Please clearly describe why the authors assume the selected models in the discussion outperformed other models. It looks clear in the fitted ROC curve model. But some other statistical analyses demonstrate homogeneities in sensitivities and specificities of different models.

4.     Figure 5B. Please keep the same color coding as Figure 5A.

Author Response

Thank you for your comments and suggestions concerning our manuscript entitled "Diagnostic Accuracy of Machine-Learning Models on Predicting Chemo-Brain in Breast Cancer Survivors Previously Treated with Chemotherapy: a Meta-Analysis"(Manuscript ID: ijerph-2068506). Those are all valuable and very helpful for revising and improving our paper. We have studied all comments carefully and have made corrections using tracked changes.

1.     In the screening of the publications, 20 articles were input but only 3 were selected. Is it too restricted the screening criterion is? Are there any articles in the excluded ones worth discussions? Though they might lack some of the elements in the result reporting.

Thank you for your observation. We reviewed again our initial articles and no important elements were added.

2.     In the literature searching step, the keyword “chemotherap” was used instead of “chemotherapy”. Is it a typo? Or “chemotherap” could function as the wildcard to pick all results with “chemotherapy” or “chemotherapeutic”? Please make double check and make sure this will not affect the expected searching results.

We agree with your observation. We used “chemotherap” to pick all results with “chemotherapy” or “chemotherapeutic”. We double checked and our search results were not modified.

3.     Please clearly describe why the authors assume the selected models in the discussion outperformed other models. It looks clear in the fitted ROC curve model. But some other statistical analyses demonstrate homogeneities in sensitivities and specificities of different models.

We demonstrated the homogeneities in sensitivities (p-value = 0.261) and specificities (p-value = 0.807) of our seven selected models – see lines 231-233. We added more discussion regarding our selected models.

4.     Figure 5B. Please keep the same color coding as Figure 5A.

We agree keeping the same color coding for crosshair plot and ROCellipse plots should be easier to read them for exploratory purposes, but we used a package in R that has predefined functions: we used the function crosshair from mada package with arbitrary coloring for crosshair plot and with grey coloring for ROCellipse plots.